# An Intelligent Hand-Assisted Diagnosis System Based on Information Fusion

**DOI:** 10.3390/s24144745

**Published:** 2024-07-22

**Authors:** Haonan Li, Yitong Zhou

**Affiliations:** Shien-Ming Wu School of Intelligent Engineering, South China University of Technology, Guangzhou 510641, China; 202120160316@mail.scut.edu.cn

**Keywords:** computer vision, knowledge graph, diagnostic system

## Abstract

This research proposes an innovative, intelligent hand-assisted diagnostic system aiming to achieve a comprehensive assessment of hand function through information fusion technology. Based on the single-vision algorithm we designed, the system can perceive and analyze the morphology and motion posture of the patient’s hands in real time. This visual perception can provide an objective data foundation and capture the continuous changes in the patient’s hand movement, thereby providing more detailed information for the assessment and providing a scientific basis for subsequent treatment plans. By introducing medical knowledge graph technology, the system integrates and analyzes medical knowledge information and combines it with a voice question-answering system, allowing patients to communicate and obtain information effectively even with limited hand function. Voice question-answering, as a subjective and convenient interaction method, greatly improves the interactivity and communication efficiency between patients and the system. In conclusion, this system holds immense potential as a highly efficient and accurate hand-assisted assessment tool, delivering enhanced diagnostic services and rehabilitation support for patients.

## 1. Introduction

Numerous neurological disorders (such as Parkinson’s disease and stroke) and musculoskeletal diseases/injuries (such as fractures, joint dislocations, joint stiffness, and hand deformities) can lead to hand function impairment, manifested as limited finger flexion and extension, hand tremors, and hand muscle weakness, which can seriously hinder patients’ normal work and life [1]. Hand function assessment plays a crucial role in diagnosing the degree of hand function impairment, optimizing treatment plans, and monitoring treatment effects.

Through the combined use of assessment scales to evaluate indicators such as hand range of motion, strength, coordination, and sensibility, rehabilitation therapists can understand the extent of the patient’s hand function impairment and effectively monitor the development of the condition, thereby formulating personalized rehabilitation treatment plans [2]. Hand function assessment is integrated throughout the diagnosis and treatment process, including injury assessment, injury cause analysis, baseline assessment before rehabilitation, progress monitoring during rehabilitation, and efficacy evaluation after rehabilitation to ensure the safety, effectiveness, and standardization of rehabilitation treatment.

Timely and accurate assessment can help patients implement individualized treatment plans, further reducing the incidence of secondary complications of hand diseases, and thereby improving patients’ self-care ability [3,4]. However, the current hand function diagnosis process is labor-intensive and time-consuming, and many patients cannot access sufficient and accurate motion measurement and functional assessment due to the shortage of professional personnel and the uneven distribution of medical resources.

The development of an intelligent hand-assisted diagnostic system can provide patients with smart, stable, and accurate diagnoses and recommendations, alleviate the burden on clinical medical personnel, and solve the problem of the shortage of professional therapists, thereby meeting the demand for diagnosis and rehabilitation training of motion function disorders [5,6,7]. However, most current research has focused on robotic devices for hand function rehabilitation training, which relies on wearable devices such as data gloves with attached sensors to complete hand sensing and recognition. This approach has high requirements for hardware such as sensors and glove materials, and the cost is significantly increased, requiring professional supervision and guidance, which is not easy to widely promote for daily use.

Compared to the complex process of installing, calibrating, and adjusting sensor-based devices, the use of machine vision for replacement can not only achieve rapid migration and application in different scenarios but also will not cause constraints on hand movement. However, research on hand function diagnosis based on computer vision is extremely limited, and there are few specialized devices for hand motion assessment on the market. Therefore, it is urgent to conduct research on the key issues of the automated assessment of the patient’s hand condition, medical knowledge graphs related to the hand, and disease diagnosis mechanisms and to develop key technologies and systems for vision-based hand function-assisted diagnosis and medical knowledge graph-based intelligent medical decision support to promote the clinical application of related technologies.

This paper will establish an intelligent hand function assessment method based on hand posture and hand movement capture, which can not only objectively and accurately assess the patient’s hand condition but also provide a reference basis for the rehabilitation plan. This is of great significance in solving the serious shortage of hand function diagnoses and improving the diagnosis and rehabilitation training effects for a wide range of hand function patients.

The principal contributions of this study can be summarized as follows:1.Collected and integrated medical data related to hand function diagnosis to construct a knowledge graph-based hand disease Q&A system. Additionally, the integration of speech recognition technology enhances the system’s applicability in various scenarios, catering to the diverse needs of patients and doctors.2.Developed a high-precision and robust hand activity recognition algorithm that meets clinical application standards. This algorithm can accurately assess the functional status of patients’ hands, providing critical support for doctors in diagnosis and treatment.3.Constructed an intelligent hand-assisted diagnosis system that integrates the aforementioned Q&A and activity recognition technologies. This system offers intelligent and precise diagnostic support for healthcare professionals.

The remainder of this paper is organized as follows: Section 2 analyzes the current state of research on hand-assisted diagnosis, summarizing the challenges and issues currently faced. Section 3 introduces the overall architecture of our proposed assistive diagnosis system, providing detailed descriptions of our hand medical Q&A model and visual algorithm model. Section 4 evaluates and compares the system’s actual performance in hand activity diagnosis and interactive Q&A tasks. Finally, Section 5 concludes the paper and discusses future work.

## 2. Related Work

### 2.1. Wearable Sensors

Compared to indirect measurements from visual signals, wearable sensors are more precise in measuring joint angles and positions. However, patients with severe hand impairments may face difficulties in wearing them, necessitating specially designed exoskeleton robots suitable for the patients. Wearable exoskeleton hand robots can be primarily categorized into two types: rigid and soft. Rigid wearable hand robots generally consist of rigid actuators connected to the drive-through cables or linkage (crank) structures, equipped with relevant sensors for system control. Some studies have employed electromyography (EMG) sensors for precise control of the exoskeleton [8], and rich physiological signals can further enhance the functionality of the device [9,10]. While others have focused on the design of hand robots, such as HEXORR [11] and HANDEXOS [12]. These devices can achieve finger flexion and extension but have their respective limitations. Not only have commercial companies designed devices with force feedback and full finger flexion and extension functions, but Chinese research teams have also made contributions in this field, such as the University of Chinese Academy of Sciences [13], Harbin Institute of Technology [14], and the University of Science and Technology of China [15]. The devices they developed have all adopted advanced sensors and controllers. Furthermore, DextaRobotics [16] and Shanghai Fourier Intelligence have also developed distinctive commercial products. However, rigid exoskeleton hand robots have the following issues: the heavy rigid finger joints and low degrees of freedom add burden to the patient’s hand, lacking a certain degree of “compliance”; they rely too much on theoretical modeling and motor encoder monitoring functions, and have not been implemented in the real bending deformation detection of finger joints, which may cause secondary injuries to the patient’s hand due to theoretical model deviations and encoder obstacles.

In contrast to rigid wearable hand robots, soft wearable hand robots are generally made of soft actuators, fabrics, or other forms of soft materials in the form of gloves, connected to electric/pneumatic actuators through threads/air chambers. This provides great flexibility and compliance, allowing the system to easily adapt to the surrounding environment and the patient, thus improving comfort, safety, and ease of use. Among them, the soft pneumatic glove developed by Harvard University [17] is designed based on the anatomical structure of human fingers, using segmented pneumatic bending actuators to ensure that the bending contour of the soft glove remains consistent with the fingers. Yap et al. [18] from the National University of Singapore achieved a precise perception of the bending motion of the hand through soft pneumatic bending actuators and flexible fiber Bragg grating (FBG) sensors. Tobias et al. [19] from the Swiss Federal Institute of Technology in Zurich used a three-layer sliding spring mechanism as the actuator for the soft exoskeleton robot, causing the exoskeleton to bend by changing the relative length of the springs. However, this set of devices lacks any sensing solution and cannot measure joint angle changes in hand assessment. Kang et al. [20] from Seoul National University in Korea developed a soft wearable robot supported by polymer materials, using cables to replace tendon tissue based on hand anatomy to drive the soft glove to bend. Kim et al. [21] from the Korea Advanced Institute of Science and Technology also designed a soft robotic glove with high degrees of freedom using a tendon-driven approach, capable of achieving 8 degrees of freedom of motion. However, these solutions do not have a sensing system. Lai et al. [22] from Southeast University introduced a hybrid actuator that combines a silicone bending actuator with a shape memory alloy (SMA) spring actuator. By changing the environmental temperature through water cooling, the SMA spring actuator undergoes displacement changes, causing the silicone bending actuator on the glove to produce flexion and extension. Chen et al. [23] from Fudan University introduced a wearable hand rehabilitation system in which the user’s healthy hand wears a data glove equipped with force and bending sensors, while the affected hand undergoes mirror therapy wearing a motion glove. The soft motion glove employs a tendon-driven approach, allowing independent control of each finger. Deng et al. [24] from Huazhong University of Science and Technology proposed a tendon-driven lightweight wearable glove called Sen-Glove, which features a soft joint sensing solution. This glove can monitor the full bending motion of 14 joints across five fingers, recognize 11 gestures and assist subjects in grasping 21 multi-scale objects. Currently, both domestic and international markets for soft wearable hand robots are based on soft pneumatic actuator design solutions combined with data gloves worn on the healthy hand to achieve functions such as mirror therapy. This solution currently suffers from low motion perception accuracy and low degrees of freedom in hand activity. However, existing soft exoskeleton hand robots have the following issues: the unavoidable haptic feedback problem in soft exoskeletons, where the “compliance” also leads to the inability to precisely obtain force feedback in the flexion and extension states of each finger joint; the irrational distribution of soft actuators may also produce significant impedance to the motion coupling of finger joints. Finally, as an essential component of robots, soft multimodal sensors for the hand have not been thoroughly investigated.

### 2.2. Visual Diagnostic Assistance

Vision, as a novel sensing modality, can perfectly address the issues associated with wearable sensors and offers the following advantages in realizing hand posture perception: low cost, stable and replaceable hardware; no restriction on hand flexibility, reducing the operational burden; no data drift, eliminating the need for complex manual calibration before use.

Visual-based pose estimation of limb movements, as a fundamental technology, has been widely applied in various scenarios such as film and television, sports, and entertainment. Microsoft’s Leap Motion [25], Intel’s Realsense [26], and Google’s Mediapipe technology [27] are leading in this field. Most products adapt open-source algorithms for application, lack targeted design, inability to customize joint points, have difficulty in improving recognition accuracy, and inability to meet the needs of hand diagnosis scenarios. Moreover, the development and validation of artificial intelligence algorithms based on clinical cases require the support of a large patient database and the reliance on medical institutions with extensive influence, resulting in a longer development cycle.

Vision-based hand function measurement has the advantage of unobtrusive measurement. However, there is limited research on vision-based hand function measurement and diagnosis, primarily focusing on post-stroke rehabilitation and diagnosis. Visual rehabilitation systems can provide patients with a time-point assessment of movement quality related to one or more body parts, thereby estimating the patient’s condition, which can be performed in clinical or non-clinical environments (such as at home). For example, Metcalf et al. [28] proposed a Kinect depth camera-based hand motion measurement method and validated its feasibility for use in a home rehabilitation setting. Zestas et al. [29] proposed a vision-based hand rehabilitation assessment suite based on the Box and Block Test (BBT) and the Sollerman Hand Function Test (SHT), utilizing existing visual techniques for hand and finger tracking, simulating assessment scenarios, and scoring. Most systems evaluate patients based on their performance in virtual reality rather than directly assessing their physical performance. For instance, NeuroTechR3 employs a vision algorithm based on infrared cameras to develop a virtual reality rehabilitation system for post-stroke hand and arm recovery. Sucar et al. [30] developed a visual gesture therapy system for rehabilitation training in a virtual reality environment. Additionally, some studies make assessments or provide feedback based on the subject’s physical performance in the physical world, with condition measurement relying on built-in algorithms or requiring the assistance of clinicians. For example, Zariffa et al. [31] used contour features extracted from background subtraction as features for classifying changes in grip and employed KNN for classification, applying it to posture recognition related to ADL neurorehabilitation and functional recovery. However, this approach has the issue of environmental dependence, especially when a part of the hand is occluded, which may impact the measurement accuracy of finger movements or joint angles.

### 2.3. Medical Question Answering System

In the current era of information interconnectedness, the widespread use of the Internet has enabled people to easily access professional medical and health knowledge from all over the country, transcending geographical and temporal limitations. However, with the increasing number of Internet users and the extensive application of big data technology, personalized recommendations and bidding advertisements have flooded web pages, making it difficult for people to distinguish between true and false information in the vast amount of medical and health data.

In this context, using search engines to obtain medical knowledge is often limited because keyword-based search results struggle to meet users’ precise and professional needs. In contrast, medical question-answering systems, through information retrieval and natural language processing techniques, can better understand the questions posed by users and provide accurate and authoritative answers. These systems leverage Internet architecture and strive to provide users with personalized and professional medical and health consultation services, far surpassing the search results of traditional search engines. With the development of information technology, knowledge graphs (KG) [32], due to their powerful semantic representation and data organization capabilities, have redefined the structure of knowledge from a new perspective. Structured graph data have provided a new direction for the development of question-answering systems, namely knowledge graph-based question-answering systems.

Currently, the coverage of question-answering systems on the market regarding hand diseases is still limited, and there is a lack of information, especially professional guidance and advice specific to hand diseases. To meet users’ concerns and needs for hand health, there is an urgent need to develop a question-answering system specifically targeted at hand diseases, providing comprehensive and professional information and guidance.

Medical question-answering systems play a crucial role in promoting subjective interaction with patients. Through medical question-answering systems, patients can ask a series of relevant questions to help obtain information about their current symptoms, functional impairments, and daily living assistance. This information gathering not only assists doctors in more accurately diagnosing the condition but also helps doctors understand the difficulties and challenges patients may face in their daily lives.

Therefore, the introduction of medical question-answering systems can significantly enhance the positive interaction between patients and the system, providing patients with more comprehensive and intelligent diagnosis and treatment support. This comprehensive assessment approach not only helps improve patient quality of life but also enhances the overall efficiency and satisfaction of hand medical assistive systems.

In summary, we have synthesized the comparative advantages and limitations of the three systems in Table 1. Based on this analysis, we propose to develop an innovative hand-detection intelligent system that synergistically integrates visual recognition technology with medical question-answering systems to enhance diagnostic accuracy and assessment capabilities for hand disorders. This integration capitalizes on the objectivity of visual recognition and the interactive nature of question-answering systems, aiming to transcend the limitations of individual technologies and achieve a more comprehensive and precise hand health evaluation. It amalgamates the non-invasive, stable, and cost-effective attributes of visual diagnostics with the capacity of medical question-answering systems to capture patients’ subjective symptomatology and quality of life impacts.

## 3. Methods

We present the design of an intelligent hand diagnosis system for medical assistance. The system architecture consists of three main components: visual perception of hand information, intelligent question-answering based on a knowledge graph, and a human–computer interaction interface.

In the knowledge graph-based intelligent question-answering module, we leverage the medical information related to hand diseases collected from large-scale websites. By applying information retrieval techniques, we analyze and extract professional medical knowledge to construct a knowledge graph database based on a graph structure. This graph-structured database is centered around hand diseases and links information about diseases, symptoms, diagnostic criteria, treatment methods, and preventive measures for different parts of the hand. By matching the keywords in user queries, relevant results can be quickly retrieved, providing corresponding medical information, or suggestions. Moreover, we incorporate a speech recognition model to assist patients in inputting their questions through simple conversational communication, facilitating the self-reporting of their condition.

In the visual perception module, we have developed a set of visual perception algorithms that utilize monocular image recognition techniques. The real-time image data captured by the camera are fed into the algorithms to recognize and understand the patient’s hand information. Our algorithms output the morphological mesh and pose of the hand in the camera coordinate system. By analyzing the relationships between the three-dimensional coordinates of these spatial points, valuable information about the hand’s geometric shape, such as bone lengths and joint angles, can be extracted. Furthermore, real-time tracking enables the analysis of temporal changes. Additionally, our module supports the integration of clinical assessment forms or evaluation tools to provide a more comprehensive understanding of hand-related information in medical applications. These assessments can be based on various factors, such as hand function, range of motion, or strength, offering valuable insights into the patient’s condition and progress over time.

In the human–computer interaction module, we provide users with a clear, user-friendly, and comprehensive interface. The patient assessment process and the final evaluation results from the previous two modules are displayed in real time.

We have constructed a comprehensive experimental system. The core layout of the system is shown in Figure 1. We have selected the Logitech C920PRO as the primary data acquisition device. This device integrates a high-quality monocular camera and a microphone, effectively capturing hand images and receiving voice input, meeting the basic data input requirements of the experiment.

During the data acquisition stage, the system processes the video by frame extraction or directly records the audio, uploading the image and voice data to the server. Subsequently, these data are fed into the specially designed visual perception algorithms and medical question-answering system for processing. The visual perception algorithms analyze the hand images to identify hand motion information and determine whether the range of motion is normal. The medical question-answering system processes the voice input and provides targeted disease information and suggestions. Finally, the processing results are fed back to the user interface in real time.

### 3.1. Establishing a Question-Answering System

#### Question-Answering Knowledge Base from Information Retrieval

We adopt the information retrieval method [33] to construct a question-answering interaction system with the knowledge graph database. First, we build a knowledge base (KB) with a graph data structure based on professional hand-related medical knowledge. Given a user-inputted question, the system determines the query intent and constructs the corresponding retrieval source. Subsequently, the system links to the topic entity $e_{q}$ in the knowledge base and extracts a set of relevant candidate answers from the corresponding knowledge subgraph. The system then extracts feature vectors from each of these candidate answers. Finally, a pre-trained ranking model is used to score and predict the candidate answers, outputting the final answer results. The specific process flow is shown in Figure 2.

After determining the topic entity $e_{q}$ in the question *q*, the system extracts a local subgraph from the constructed knowledge graph G that is specific to the question. Ideally, this subgraph will cover all entities and their relationships relevant to the question, represented as nodes and edges, respectively. This graph-structured knowledge representation approach allows for reasoning and computation on the graph using information retrieval techniques without explicitly generating executable logical forms, ultimately deriving the final answer results. We introduce a retrieval source construction module that takes both the question and the knowledge base as inputs, Equation (Equation 1):(1)$$G_{q}=R\_S\_C\left(q,G\right),$$
where $G_{q}$ is the question-specific graph extracted from G.

Next, the system semantically encodes the input question through the Question Representation module. This module analyzes the semantic features of the question and transforms them into reasoning instructions, which are typically represented as vectors, as shown in Equation (Equation 2):(2)$$i^{k}=Q\text{\_}R\left(i^{k-1},q,G_{q}\right),$$

Here, $i^{k},k=1,\ldots ,n$ is the *k*-th reasoning instruction vector, used to encode the semantic and syntactic information of natural language questions. Both multi-step reasoning and one-step matching are applicable, resulting in different reasoning steps *n*.

The graph-based reasoning module (Graph-Based Reasoning) performs semantic matching through vector-based computations, propagating and aggregating information along adjacent entities within the graph. The reasoning state $s^{k},k=1,\ldots ,n$ is updated according to the reasoning instructions, which can be represented by Equation (Equation 3):(3)$$s^{k}=Q\text{\_}B\text{\_}R\left(s^{k-1},i^{k},G_{q}\right),$$
where $s^{k}$ is the reasoning state, representing the state of the *k*-th reasoning step on the graph.

The answer generation module (Answer Generation) generates answers based on the reasoning state upon the completion of reasoning. We use an entity ranking generator to rank entities, obtaining the top-ranked entities as predicted answers. This module can be formalized as Equation (Equation 4):(4)$${\tilde{A}}_{q}=A\text{\_}G\left(s^{n},G_{q}\right),$$
where $s^{n}$ represents the reasoning state at the final step. The entities contained in $G_{q}$ are the candidate entities for the answer prediction ${\tilde{A}}_{q}$. In many cases, ${\tilde{A}}_{q}$ is obtained by selecting entities with scores higher than a predefined threshold, where the scores are derived from $s^{n}$.

### 3.2. Knowledge Graph Construction for Question-Answering System

#### 3.2.1. Data Collection and Graph Construction

To provide more professional medical advice related to hands, we leverage the semi-structured data from the [34] website to improve the quality of the knowledge graph. The data collection process is primarily carried out using Python web crawling modules. The data we collected was in Chinese. In order to facilitate understanding, we have replaced the original Chinese content in the following sections with its English translation. Following the categorization of the website, we collect the following data items: introduction, causes, prevention, complications, symptoms, examination methods, treatment methods, drug recommendations, and dietary care. To constrain the scope of diseases to hand-related conditions, we select data containing the ”hand” in the introduction, causes, complications, and symptom categories, excluding the term ”surgery”. This results in a curated dataset of 1805 hand-related diseases, covering information such as disease names, incidence rates, susceptible populations, transmission routes, treatment methods, treatment duration, cure rates, drug details, drug recommendations, dietary restrictions, recommended foods, symptoms, causes, preventive measures, disease categories, commonly used drugs, and complications. The cleaned data are then stored in a local MongoDB database for persistence.

During the construction of the knowledge graph, we utilize the Python programming language and the Neo4j graph database tool. By following the Cypher query language construction method, we import the locally stored data into the Neo4j database. The determination of entities and relationships in the graph primarily relies on the category labels provided by the website, which are then formatted into label types. The resulting knowledge graph consists of 4494 entities and 20,828 corresponding relationships. We summarize the attribute types, entity types, and entity relationship types in Table 2, Table 3, and Table 4, respectively. In our knowledge graph sample, there is significant coverage of food-related instances, which is mainly influenced by the rich dietary advice content available on the [34] website. The reliability of some data remains debatable, and in the next step, we plan to introduce more authoritative data sources for further adjustment and optimization.

We have found that the number of medication entities in our collected database is relatively small, which may be attributed to the diversified, specialized, and complex nature of hand therapy. Generally, hand disease treatment may not be limited to medication but also includes physical therapy, surgical interventions, and rehabilitation training. Common rehabilitation methods may place more emphasis on non-pharmacological approaches, leading to a relatively small number of medication entities. Some hand diseases may not have clear medication treatment protocols or may require limited medication options. For example, certain types of arthritis may require medication, but the selection of medications may be limited. Some hand diseases may require specific professional knowledge and diagnosis, and treatment plans may need to be individually tailored. For certain hand diseases, physicians may be more inclined to recommend non-pharmacological treatment methods, which may be considered more effective or safer, thereby reducing the need for medication and the number of medication entities.

#### 3.2.2. Question Classification

We have designed a simple question classifier to automatically determine the topic and entity types involved in a user’s question by analyzing the question itself. This allows the system to better understand the user’s intent and provide appropriate responses. The question classifier works through the following steps.

Initialization: During the program initialization phase, keyword dictionaries containing various medical entities (such as diseases, foods, drugs, etc.) are loaded based on our dataset. At the same time, an Aho-Corasick automaton is constructed to create an automaton tree for fast keyword matching, which is more efficient than simple string matching.

Question Analysis: Based on the semantics and entity information in the user’s question, we classify the question into different subcategories. These subcategories may include disease symptom queries, disease cause queries, food recommendation queries, and so on. By identifying specific keywords or key phrases in the question, such as symptoms, causes, foods, etc., the topic and type of the question can be determined for classification purposes.

Classification: The classifier categorizes the question into the appropriate subcategory based on its type and returns it to the subsequent processing component. This allows the subsequent question parser to generate corresponding query statements based on the question type, searching for relevant data in the medical knowledge graph to meet the user’s needs.

Through the processing of the question classifier component, the system can effectively identify the topic and type of the user’s question, providing a foundation for subsequent query processing. In this way, the system can more accurately understand the user’s intent and provide relevant medical information or suggestions accordingly.

#### 3.2.3. Knowledge Graph Matching and Resulting Feedback

In the knowledge graph matching and resulting feedback stage, we retrieve relevant information from the medical knowledge graph based on the question type determined by the question classifier and return the information to the user in a readable format. This enables users to obtain the required medical information or advice, thereby better understanding their health issues and taking corresponding measures.

After the question classifier determines the question type, we construct an entity dictionary based on the returned entity information. The dictionary uses entity types as keys and corresponding entity lists as values. Meanwhile, for each question type, we generate a query statement according to the corresponding processing logic. For different question types, we use specific query templates to generate corresponding SQL query statements. Based on the question type and corresponding entity information, we generate the query statement. The purpose of the query statement is to retrieve relevant information from the knowledge graph. We send the generated query statement to the knowledge graph database for execution. The knowledge graph database retrieves relevant information from the stored data based on the query statement. The query results are returned to the user in a structured format. These results may include relevant information about diseases, symptom descriptions, medication recommendations, and so on, depending on the question type posed by the user.

#### 3.2.4. Interactive UI Development

To achieve a superior user interaction experience, we have developed a UI interface using HTML, CSS, and JavaScript, as shown in Figure 3. This UI system excels in terms of intuitive usability. Its clean and straightforward interface design and intuitive button layout enable users to quickly understand and operate the system, reducing the difficulty and error rate of user operations and improving the quality of the user experience. Secondly, user-friendly interaction is one of the notable features of this UI system. By presenting the conversation content in a dialog box format, users feel as if they are communicating with a real customer service representative, enhancing the naturalness and friendliness of the interaction. Moreover, the UI interface design takes into account user habits, providing clear input prompts and empty content prompts, making user operations more smooth and convenient. The responsive design for different screen sizes ensures a good user experience on various devices, increasing the flexibility and versatility of the system.

Patients with hand diseases often experience difficulties in writing or typing due to hand function impairment, which poses certain obstacles when interacting with a question-answering system. To reduce the communication cost for hand disease patients when interacting with the question-answering system, introducing speech recognition technology is an excellent solution. We have chosen the open-source Chinese model ASRT (Auto Speech Recognition Tool), which uses a convolutional neural network (CNN) and connectionist temporal classification (CTC) method to convert sound into a sequence of phonemes and then uses a pre-trained language model to transform the phonemes into meaningful Chinese. However, during testing, we found that the existing language model was not satisfactory in the specialized hand medical scenario. Therefore, we added training data to the existing model to enhance the language model’s dictionary. We specifically added vocabulary related to hand treatment, which greatly improved the system’s performance in hand medical communication scenarios.

### 3.3. Establishing a Visual System

The algorithm primarily adopts our previously proposed deep learning-based single-camera model [35], with some supplementary optimization.

#### 3.3.1. Algorithm Optimization

To achieve the goal of hand diagnosis, our algorithm relies on visual prediction to detect hand activity by estimating hand joint angles. To improve the recognition accuracy of joint angle changes, we introduced a novel consistency loss term, namely hand joint surface normal consistency, into the visual algorithm. Specifically, we consider every three connected hand joints as a joint surface and require these joint surfaces to maintain high normal consistency:(5)$$L_{n}=\sum_{f\in F}\sum_{(i,j)\subset f}\left|\frac{{\hat{V}}_{i}^{3D}-{\hat{V}}_{j}^{3D}}{\left|\right|{\hat{V}}_{i}^{3D}-{\hat{V}}_{j}^{3D}{\left|\right|}_{2}}\cdot n_{f}^{\left(gt\right)}\right|,$$
where *f* is the index of the hand joint surface F, (i,j) represents the indices of the vertices forming the edges of the triangle *f*, $n_{f}^{gt}$ is the ground-truth normal vector of the triangle face *f*, calculated from the ground-truth 3D vertices $V^{3D(gt)}$, and ${\hat{V}}^{3D}$ represents the predicted 3D joint coordinates by our algorithm. This can better capture the coordination and continuity of hand joints during activity, thereby enhancing the reliability of activity detection. After introducing the loss function, we fine-tuned and optimized the model on the FreiHAND dataset [36], ultimately obtaining the basic model.

#### 3.3.2. Algorithm Extension

To better complement our algorithm model in achieving hand prediction functionality in the target scenario, we incorporated the YOLO algorithm [37] as a supplement. The YOLO algorithm can recognize and extract the hand region of interest from background images, meeting our requirements for real-time performance and accuracy.

To improve the algorithm’s performance, we created a dataset and fine-tuned the algorithm. First, we collected a large amount of hand image data, ensuring the diversity of the dataset to cover different scenarios, postures, and lighting conditions. These images underwent strict cleaning and annotation processing to ensure the quality of the dataset. In this step, we also augmented some images to increase the scale and richness of the dataset, thereby enhancing the algorithm’s robustness.

We fine-tuned the YOLOv8s model using our dataset, and throughout the process, we continually debugged and validated the algorithm to ensure its performance met our expectations.

## 4. Hand Mobility Test

### 4.1. Hand Kinematics Analysis

The human hand movement system is a complex biomechanical system comprising bones, joints, muscles, and tendons. The bones of the fingers, palm, and forearm provide the foundational structure for hand movements, while the connections between various joints allow for multiple degrees and directions of freedom. The physiological movement behavior of a normal human hand is primarily constrained by the bones and joints, including the wrist, fingers, wrist joint, and individual finger joints. These structures enable the hand to perform fine and precise movements, such as grasping, manipulating objects, and expressing emotions through gestures. The muscles and tendons of the hand also play a crucial role, providing the necessary force and stability for hand movements. However, the majority of specific movement patterns and ranges are determined by the bones and joints. Therefore, we focus on the positions of the bones and joints in the human hand, as illustrated in Figure 4.

The skeletal structure of the human hand is primarily composed of three parts: the phalanges, metacarpals, and carpals [38]. The phalanges are the main bones of the fingers and are critical for determining the range of finger movements. There are 14 phalanges in total, distributed among the thumb, index finger, middle finger, ring finger, and little finger. The unique shape and structure of the phalanges enable a variety of finger movements, including gripping, pressing, and manipulating objects. The phalanges can generally be categorized based on their distance from the wrist into proximal phalanges, middle phalanges, and distal phalanges, with the thumb lacking a middle phalanx.

The metacarpals consist of five bones that connect the proximal phalanges to the carpals. These bones form the supportive and cushioning structure of the palm. The position and shape of the metacarpals give the palm its concave and convex contours, providing support and cushioning. The connection between the metacarpals and the phalanges enables the human hand movement system to perform various activities, including gripping, pressing, and manipulating objects.

The carpal bones serve as the connectors between the forearm and the palm, organized into two rows with four bones in each row. Specifically, the proximal row includes the scaphoid, lunate, triquetrum, and pisiform bones, while the distal row consists of the trapezium, trapezoid, capitate, and hamate bones. These eight carpal bones are interconnected by ligaments to form an integrated structure with a very limited range of motion in daily activities. Consequently, the impact of the carpal bones on the overall mobility of the hand can be considered negligible, and this paper does not delve into their detailed discussion.

Joints serve as the bridges connecting bones, playing a crucial role in linking and supporting bone movement. Each joint possesses distinct structural and functional characteristics. Similar to the classification of bones, joints are generally categorized into interphalangeal joints (IP), metacarpophalangeal joints (MCP), and intercarpal joints.

Interphalangeal joints are classified based on the arrangement of the phalanges and are further divided into distal interphalangeal joints (DIP) and proximal interphalangeal joints (PIP) according to their relative positions. In a normal right (or left) hand, the thumb has only one PIP joint, while the index, middle, ring, and little fingers each have one DIP joint and one PIP joint, totaling nine interphalangeal joints. These joints are standard hinge joints, capable only of flexion and extension movements. Due to the constraints imposed by the flexor tendons and ligaments, the range of flexion is greater than that of extension.

The metacarpophalangeal joints connect the phalanges to the metacarpal bones, with each finger corresponding to one MCP joint. The thumb’s MCP joint is unique, being a saddle joint with a greater range of motion, allowing for flexion, extension, abduction, adduction, circumduction, and opposition movements. In contrast, the MCP joints of the other fingers are condyloid joints with a smaller range of motion, primarily facilitating flexion, extension, abduction, adduction, and circumduction movements. The high mobility of the thumb’s MCP joint enables the thumb to oppose the other fingers, allowing for complex actions such as grasping, holding, and rotating objects.

The wrist joint is the articulation connecting the hand to the forearm. It is composed of 5 carpal bones, 8 carpals, and the distal ends of the ulna and radius, totaling 15 bones. Although the wrist joint involves numerous articular surfaces, its range of motion is relatively limited, and its impact on the overall functional capabilities of the hand can be considered negligible. Therefore, in the context of this study, the wrist joint is assumed to be in a neutral, straight position.

### 4.2. Standard Evaluation of Hand Movements

To facilitate the practical measurement of the range of motion in the various joints of the hand, while aligning with our computer vision algorithms, we have divided the hand into 21 key joint landmarks. This includes a separate node for the carpal bones, and four nodes per finger, including the fingertip. By establishing a three-dimensional coordinate system on the hand, we can obtain the 3D coordinate data for these 21 key points. Based on the dot product algorithm, we can then calculate the angular motion of each finger joint, as in Figure 5b.

Let the 3D coordinates of the three key points X, Y, and Z in the hand movement be $\left(a_{i},b_{i},c_{i}\right)$, $\left(a_{j},b_{j},c_{j}\right)$ and $\left(a_{k},b_{k},c_{k}\right)$, respectively. These coordinates form the vectors $\vec{XY}$ and $\vec{YZ}$. The corresponding formula for calculating the joint angle ∠XYZ in radians is shown in Equation (Equation 6):(6)$$∠XYZ=arccos(\frac{\vec{XY}\vec{YZ}}{|\vec{XY}||\vec{YZ}|}),$$

According to the standards for upper limb functional assessment established by the Hand Surgery Branch of the Medical Association [39], the normal range of motion for the metacarpophalangeal (MCP) joints is between 70° and 90° for flexion and extension, the normal range for the proximal interphalangeal (PIP) joints is between 80° and 100°, and the normal range for the distal interphalangeal (DIP) joints is between 30° and 45°. Furthermore, the assessment criteria for hand motor function stipulate that the thumb should be able to perform opposition, and the total range of motion for the thumb’s MCP and interphalangeal joints should be greater than 90°.

Our system will use these conditions as the preliminary evaluation criteria, as detailed in Table 5. In the future, the system will incorporate additional assessment scales, such as the Fugl-Meyer Assessment Scale [40], for a more comprehensive evaluation. Based on visual analysis to assess the types and severity of hand disabilities, standardized rehabilitation plans will be developed for different hand conditions (such as fractures, muscle injuries, and joint diseases) in collaboration with hand surgeons and therapists. The system will match appropriate treatment plans and utilize computer vision technology to monitor the rehabilitation process, providing real-time feedback and long-term data for medical reference. Remote physician intervention will be supported to adjust treatment plans or pause exercises in special circumstances. By integrating this module, our intelligent system aims to comprehensively facilitate the rehabilitation journey of patients with hand disabilities, increasing efficiency and quality and reducing the workload of healthcare professionals.

### 4.3. Single-Frame Image Hand Movement Reliability Test

In our experiment for hand image acquisition, participants were instructed to sit upright in front of a table and chair, with their arms raised to position their palms at an angle of approximately 30 degrees upward relative to the camera. They were also asked to maintain a distance of 20–70 cm from the camera, as detailed in Figure 6a.

To assess the accuracy of single-frame image measurements in capturing angular variations, we conducted an experiment focusing on 14 hand joints in both extended and flexed states. A traditional goniometer served as the standard for comparison, with measurements performed according to the dorsal hand technique recommended by the American Society for Surgery of the Hand (ASSH) [41]. The experimental setup is depicted in Figure 7.

Fifteen repetitions of diverse joint configurations were randomly captured for each subject, ensuring consistency between finger posture and camera perspective. These measurements served as ground truth data, while simultaneously recording the corresponding predictions from our algorithmic model. The average angular error across the 15 repetitions was calculated to represent the final evaluation metric. The average angular errors are summarized in Table 6.

Our empirical results demonstrate that single-frame image analysis performs well in recognizing hand angles in the extended state. The maximum prediction error was 5.3 degrees, the minimum was 2.7 degrees, and the average prediction error was approximately 3.97 degrees. However, in the flexed state, our algorithm’s performance was less consistent, with a maximum prediction error of 29.7 degrees, a minimum of 6.3 degrees, and an average of 9.0 degrees.

During testing, we observed frequent spatial localization drift in the distal interphalangeal (DIP) joint of the thumb when flexed, leading to a larger prediction error for this joint. Similar instability has also been observed in Google’s MediaPipe, a monocular pose estimation algorithm. We hypothesize two potential causes for this issue:Occlusion and Complexity: In complex scenarios involving occlusion, the algorithm may rely heavily on prior models for thumb localization, introducing a degree of arbitrariness and uncertainty.Unique Thumb Biomechanics: The thumb possesses a wider range of motion and a distinct joint structure compared to other fingers. Its movement trajectory and posture variations differ significantly, potentially hindering the algorithm’s ability to fully learn and comprehend its unique motion characteristics.

To further investigate this issue, we conducted two comparative experiments focusing on the thumb’s DIP joint flexion, employing two distinct camera perspectives different from the main experiment, considering the inherent angular errors observed in the thumb. In these experiments, the camera was positioned parallel to the spatial plane formed by the thumb’s joints, capturing images from both palmar and dorsal viewpoints, as illustrated in Figure 6.

To further investigate the impact of viewpoint on thumb DIP flexion angle accuracy, we conducted additional experiments using the same measurement techniques. For each comparative viewpoint (“Comparative View 1” and “Comparative View 2”), we randomly sampled 15 instances of thumb DIP flexion.

Comparative results presented in Table 7 demonstrate that effectively reducing occlusion can improve the accuracy of thumb DIP joint recognition. However, even with optimized viewpoints, the average angular error for the thumb in flexion remained relatively high at 13.1 degrees, representing the least accurate joint estimation compared to other finger joints. Therefore, further optimization specifically targeting the thumb is necessary.

Based on the findings in [42], the accuracy of our system in real-time single-frame estimation of joint extension angles and metacarpophalangeal (MCP) joint flexion angles generally meets clinical requirements. However, the persistent challenge with thumb DIP flexion accuracy highlights the need for specialized refinements to improve overall system performance.

### 4.4. Sequential Frame Acquisition Hand Movement Reliability Test

While single-frame images have limitations in capturing dynamic temporal changes, utilizing sequential frames in video format can effectively address this shortcoming. By capturing a video sequence of continuous hand movements, we can directly observe the dynamic changes exhibited by the subject during activity. This approach provides a more comprehensive understanding of the subject’s range of motion and allows for a more accurate assessment of their motor performance.

To validate this approach, we recruited a balanced sample of nine healthy subjects, comprising six males and three females, ensuring representativeness and reliability of the results. Each participant underwent a rigorous selection process to confirm their health status and suitability for the motor tests. Each subject performed five trials, with each trial consisting of 3–4 repetitions of a continuous hand movement from an open palm to a clenched fist, as illustrated in Figure 8. Throughout the test, we recorded the angular variations of each hand joint, using the range between the maximum and minimum values as a measure of the range of motion. Thumb opposition was assessed separately, with a contact error margin of 3 cm between the thumb tip and the little finger MCP joint. Distances below 3 cm were considered successful oppositions.

To facilitate a clear understanding and visualization of the data distribution, we compiled the data into box plots, as shown in Figure 9. Box plots effectively illustrate the median, quartiles, and potential outliers, providing insights into data distribution and variability. As evident from the box plots, the predicted values from our system consistently exceed the minimum healthy activity standards. We further analyzed the actual scores for a more in-depth evaluation.

Analysis of the sequential frame data revealed perfect scores for all thumb opposition assessments. The remaining range of motion assessments were scored according to the standard criteria outlined in Table 5. The resulting scores are visualized as box plots in Figure 10. Overall, the subjects’ scores consistently fell within the highest tier of the activity standards, indicating a healthy range of motion.

These results demonstrate that our system, utilizing sequential frame acquisition, can effectively assess and identify a healthy hand range of motion.

While the above results demonstrate the superiority of our system, we acknowledge certain limitations in our research. Considering that adding markers or other auxiliary tools may affect the recognition accuracy of the monocular algorithm, we did not conduct comparative experiments with stereo devices [43]. In future research, while continuing to improve the recognition capability of the algorithm, we plan to validate and compare the integration with stereo devices.

### 4.5. Reliability Testing of Interactive Question-Answering System

Given that our proposed comprehensive question-answering system utilizes the speech recognition functionality of an open-source model for speech-to-text supplementation, we will forego functional testing of the speech component. Instead, we will focus exclusively on testing the reliability of the interactive question-answering system presented in this paper.

To ensure the system’s ability to accurately answer a variety of user questions and adapt to different input formats and sentence structures, we have collected a set of medical questions related to hand conditions. These questions cover different topics, language styles, and sentence structures, ensuring that they encompass various aspects of the entities, such as introduction, etiology, preventive measures, and treatment approaches. We have conducted tests on the interactive interface, utilizing both voice and question-answering systems. As shown in Figure 11, we have performed a comprehensive test on the question-answering system, covering areas such as etiology and symptoms, preventive measures, dietary recommendations, complications, and treatment methods. The results indicate that the system has demonstrated satisfactory performance in each of the test domains, reflecting its strong understanding and application of hand-related medical knowledge.

Particularly in the etiology and symptoms domain, the system has exhibited excellent question-answering capabilities, accurately describing the various possible causes and related symptoms and providing users with clear and informative responses. Regarding preventive measures, the system offers comprehensive and effective recommendations, helping users understand how to prevent the occurrence of hand-related diseases. In the dietary recommendation aspect, the system can provide basic dietary suggestions, guiding users toward more nutritious dietary choices. Furthermore, in the areas of complications and treatment methods, the system has also shown commendable performance. It can accurately identify potential complications and provide corresponding preventive and treatment recommendations. Additionally, for different diseases, the system offers a diverse range of treatment options, explaining the applicability and effectiveness of various treatment methods and providing users with scientifically reliable medical guidance.

In summary, through extensive testing, we have found that the system demonstrates ideal performance in different aspects of question answering, exhibiting a high level of reliability and practicality.

## 5. Conclusions

This study proposes an intelligent hand-assisted diagnostic system that integrates visual perception and cognitive interaction technologies. The visual basic framework was optimized, and the YOLO algorithm was combined to achieve real-time capture of hand motion status. A knowledge graph of hand diseases was established to support the question-answering system. The Aho-Corasick automaton was adopted to match keyword dictionaries, and speech recognition algorithms were introduced to enhance user experience. The system demonstrated reliability in various test scenarios. In the future, the integration of large-scale language models is considered to improve the system’s knowledge coverage and the accuracy of question answering, thereby enhancing its intelligence level. Additionally, the research team plans to collaborate with the rehabilitation department of hospitals to introduce the system into clinical practice, providing more comprehensive assisted diagnostic services for patients.

## Figures and Tables

**Figure 1 sensors-24-04745-f001:**
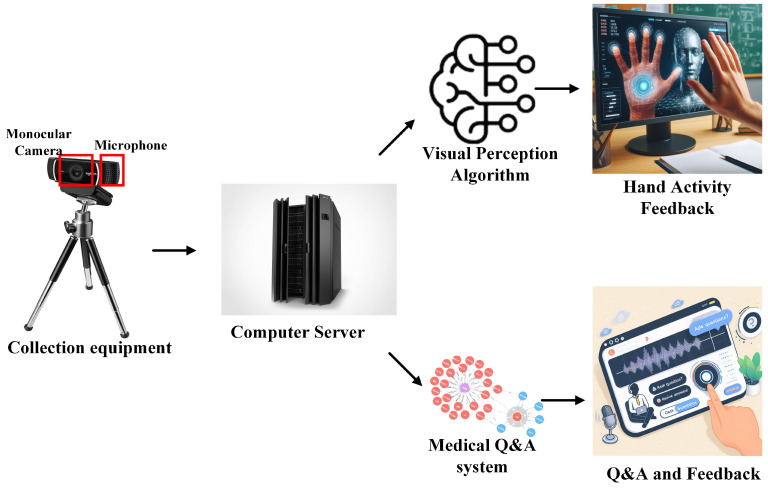
Experimental system layout.

**Figure 2 sensors-24-04745-f002:**
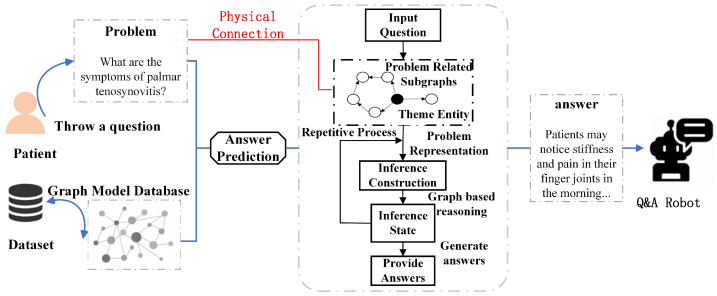
The architecture of a knowledge base question-answering system based on information retrieval.

**Figure 3 sensors-24-04745-f003:**
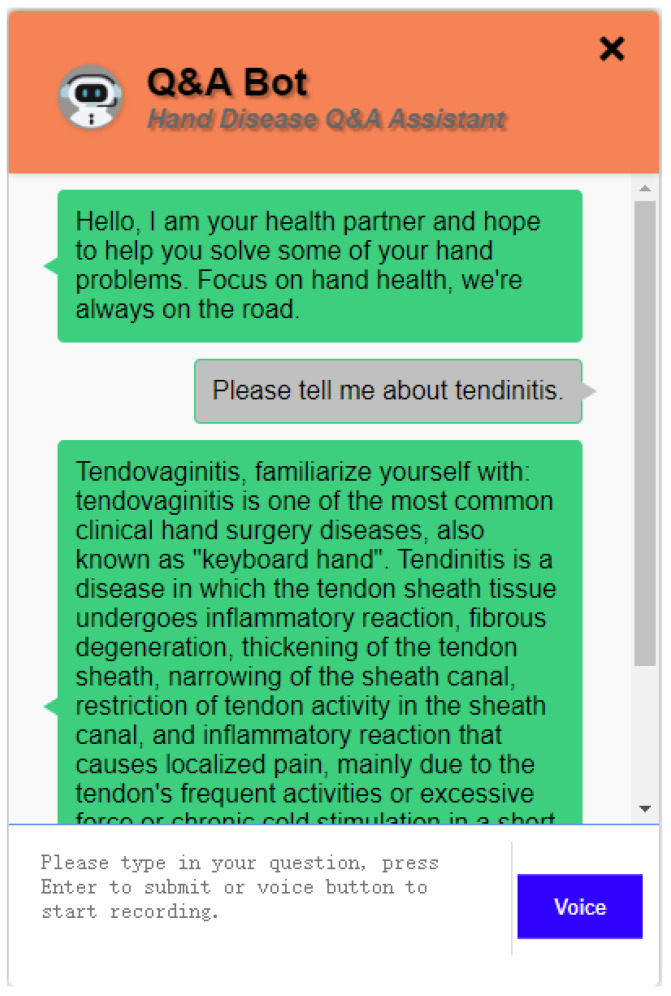
Example of UI interface for Q&A robot.

**Figure 4 sensors-24-04745-f004:**
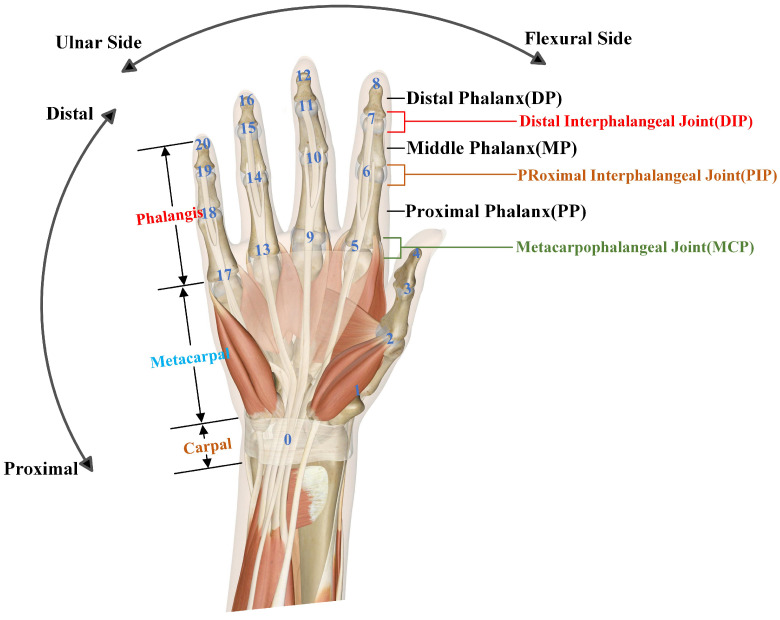
Schematic diagram of the bone and joint distribution in the right human hand.

**Figure 5 sensors-24-04745-f005:**
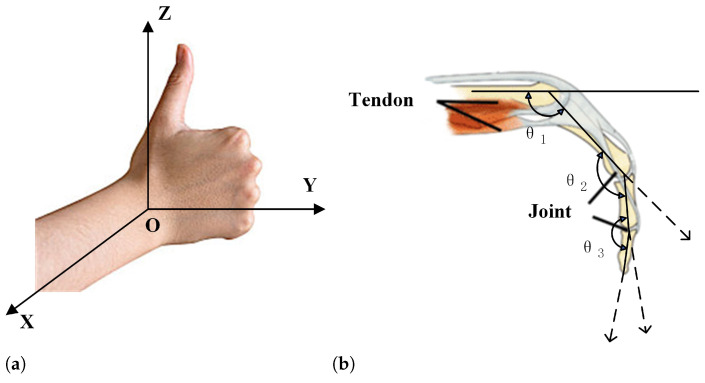
Diagram of the three-dimensional finger coordinate system and joint angles. (**a**) Example of a three-dimensional coordinate system of the hand. (**b**) Two-dimensional example of joint angles.

**Figure 6 sensors-24-04745-f006:**
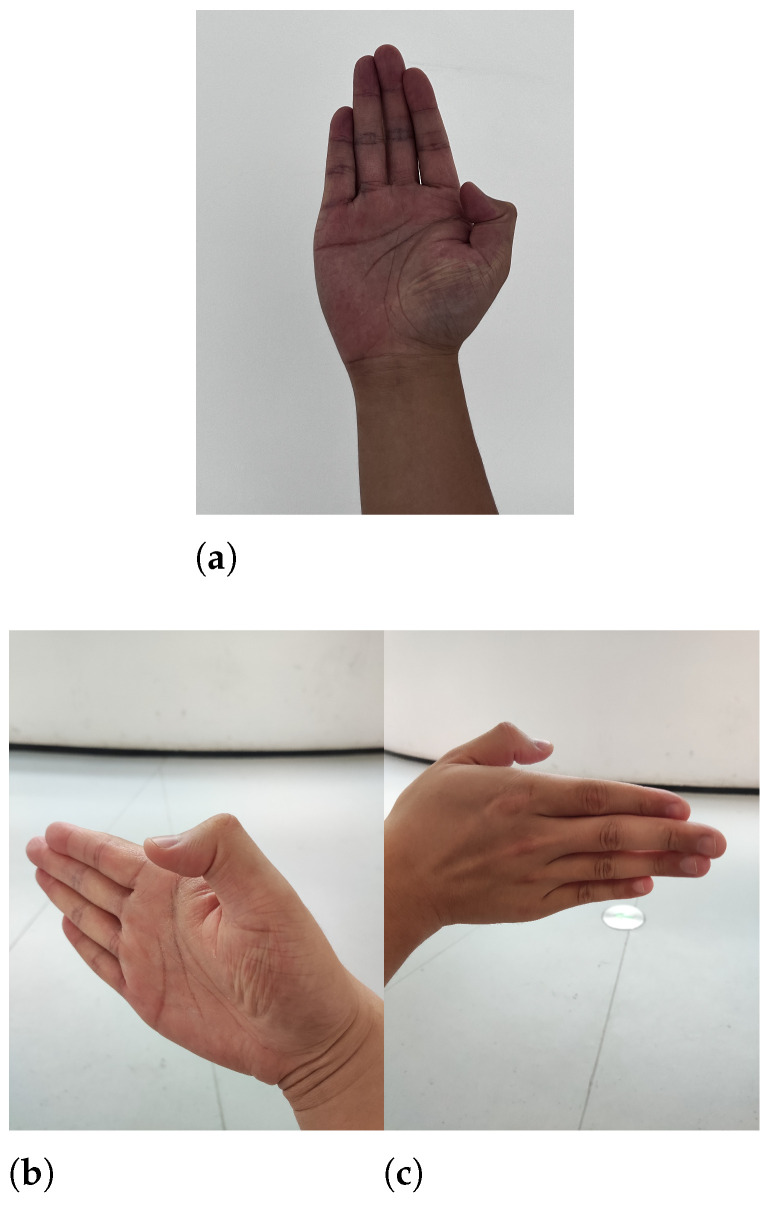
Comparative experiment shooting angles. (**a**) Main experiment viewpoint example. (**b**) Viewpoint 1. (**c**) Viewpoint 2.

**Figure 7 sensors-24-04745-f007:**
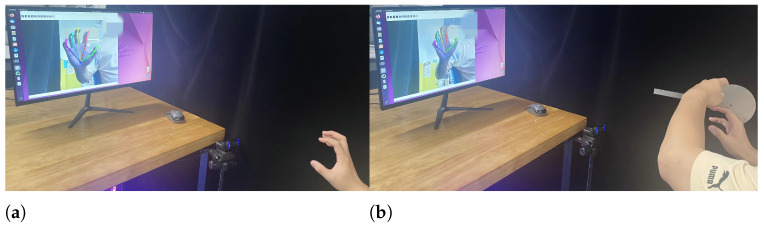
Experimental setup for single-frame image measurement. (**a**) Example of single-frame image measurement. (**b**) Dorsal hand measurement example.

**Figure 8 sensors-24-04745-f008:**
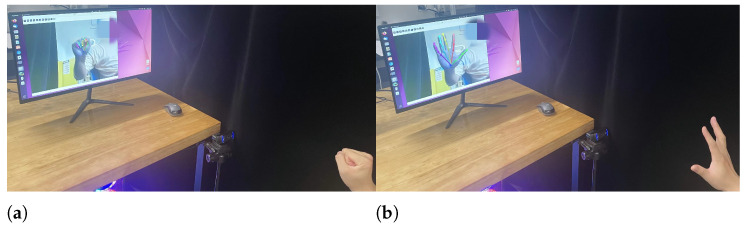
Experimental setup for continuous-frame hand motion measurement. (**a**) Hand Clenching Example. (**b**) Hand Relaxation Example.

**Figure 9 sensors-24-04745-f009:**
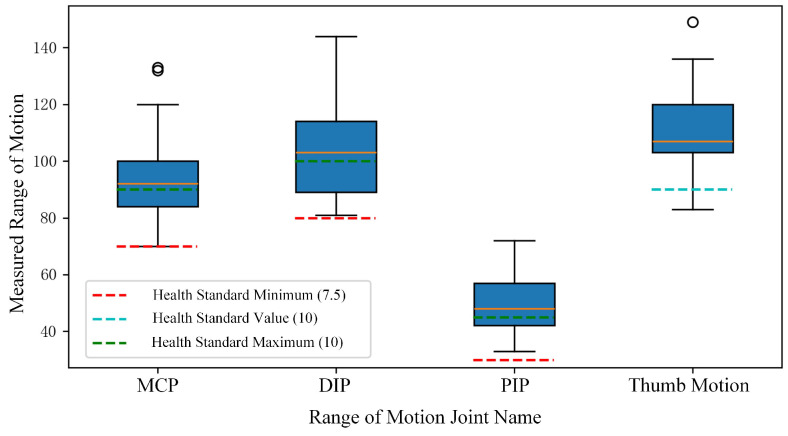
Boxplot of continuous-frame hand activity range data.

**Figure 10 sensors-24-04745-f010:**
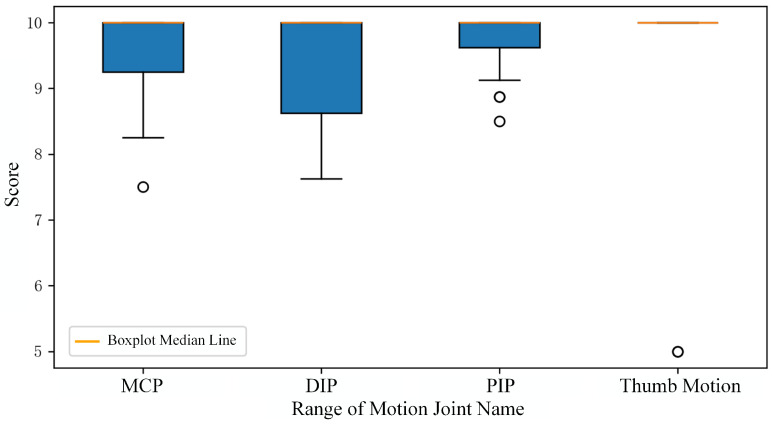
Boxplot of hand joint functional activity score.

**Figure 11 sensors-24-04745-f011:**
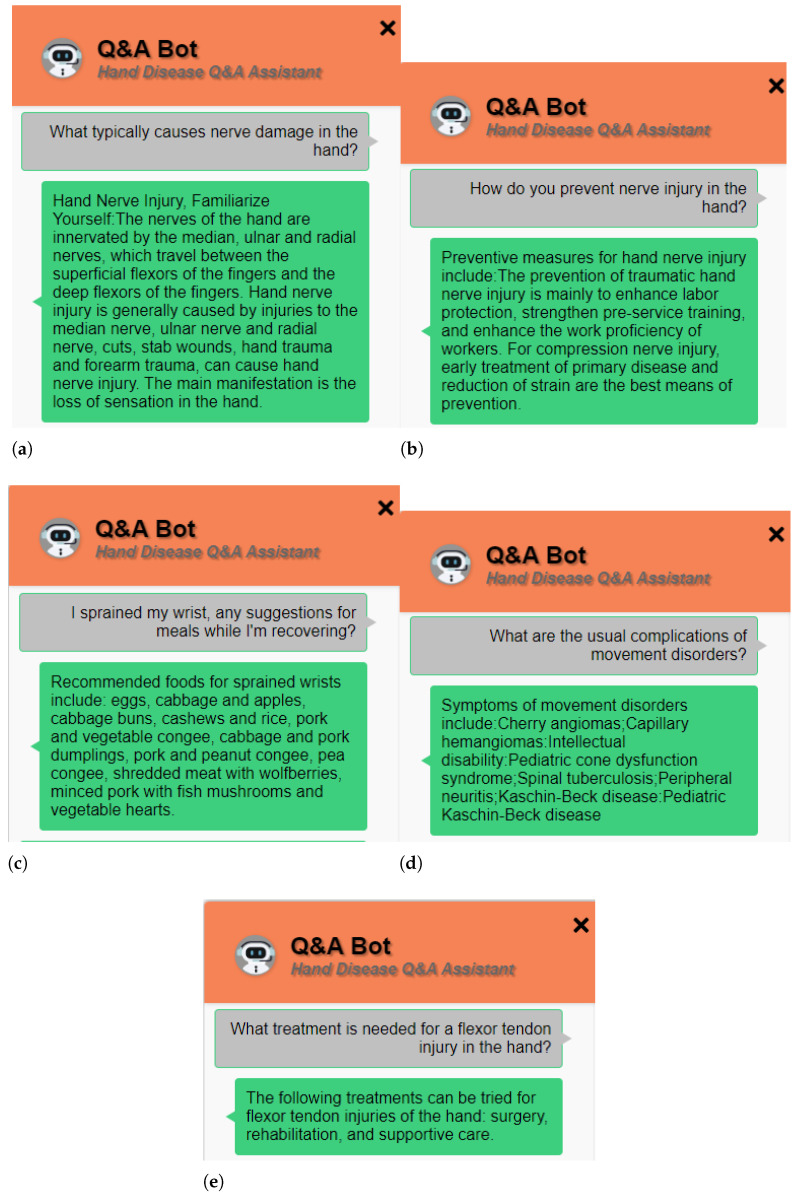
Presentation of results from experiment on reliability of question answering. (**a**) Etiology and symptoms. (**b**) Preventive measures. (**c**) Dietary recommendations. (**d**) Complications. (**e**) Treatment methods.

**Table 1 sensors-24-04745-t001:** Comparison of different methods for hand assessment and disease management.

Method	Advantages	Limitations
Wearable Sensors	Precise measurement of joint angles and positions	Not suitable for patients with severe hand impairments; fixed pattern, unable to adapt to multiple situations
Visual Diagnostic Assistance	Low-cost, stable, no restriction on hand flexibility, no data drift	Lack of targeted design for clinical scenarios; self-occlusion and other issues may lead to a decrease in accuracy
Medical Question-Answering System	More precise in answering professional medical questions than search engines; aids subjective understanding of patient symptoms and living impairments	Currently limited coverage for hand diseases

**Table 2 sensors-24-04745-t002:** Knowledge graph attribute types.

Entity Type	Meaning	Example
name	Disease Name	Tendon sheath cyst
desc	Disease Introduction	Tendon sheath cyst refers to an internal adhesive that occurs near ...
cause	Disease Etiology	The disease is more common in the back of the wrist and foot ...
prevention	Preventive Measures	Tendon sheath cysts pay attention to rest in the affected area ...
cure_lasttime	Treatment Cycle	1–3 weeks
cure_way	Treatment Methods	Surgical therapy, rehabilitation therapy, supportive therapy
cured_probe	Cure Probability	80%
easy_get	Susceptible Population	More common in young and middle-aged people

**Table 3 sensors-24-04745-t003:** Knowledge graph entity types.

Entity Type	Entity Quantity	Example
Department	51	Rehabilitation department; dermatology
Disease	1805	Compression of the dorsal scapular nerve; tendon sheath cyst
Drug	10	Da Yue Jing Wan; dexamethasone sodium phosphate injection
Food	2606	Stewed lamb with persimmons; tofu and seafood soup
Producer	22	Cephalosporin; levofloxacin tablets
Total	4494	Approximately 45,000 entity level

**Table 4 sensors-24-04745-t004:** Knowledge graph entity relationship types.

Entity Type	Entity Quantity	Example
acompany_with	1063	<tenosynovitis, comorbidities, purulent dactylitis>
belongs_to	1840	<Renault’s disease, belongs to, rheumatic immune disease>
do_eat	4686	<Hand flexor tendon injury, recommended to eat, cashew>
drugs_of	22	<Renqing Mangjue, on sale, Ganlu Renqing Mangjue>
no_eat	4692	<Achilles tendinitis, avoid eating, chicken wings>
recommand_drug	10	<Thallium poisoning, recommended medication, renqingmangjue>
recommand_eat	22	<Recommended recipe for distal fracture of torso bone, fried eggplant with egg>
Total	20,828	Approximately 21,000 entity level

**Table 5 sensors-24-04745-t005:** Hand joint functional activity standards.

Activity Name	Activity Standard	Score
MCP Flexion and Extension	70°–90°	7.5~10
	50°–69°	5~7.5
	30°–49°	2.5~5
	<30°	0~2.5
PIP Flexion and Extension	80°–100°	7.5~10
	60°–79°	5~7.5
	30°–59°	2.5~5
	<30°	0~2.5
DIP Flexion and Extension	30°–45°	7.5~10
	20°–29°	5~7.5
	15°–19°	2.5~5
	<15°	0~2.5
Thumb Opposition	can	10
	hard	5
	can’t	0
Thumb Active Range of Motion (AROM)	>90°	10
	<90°	5
	Stiffness	0

**Table 6 sensors-24-04745-t006:** Single-frame hand angle evaluation results (mean error angle, unit: degrees).

Activity State	Thumb DIP	Thumb MCP	Index DIP	Index PIP	Index MCP
Extension	5.3	3.7	2.8	3.3	4.1
Flexion	29.7	4.8	7.8	6.8	8.0
**Activity State**	**Middle DIP**	**Middle PIP**	**Middle MCP**	**Ring DIP**	**Ring PIP**
Extension	3.8	4.8	5.2	5.1	3.6
Flexion	7.0	7.8	7.5	5.9	5.6
**Activity State**	**Ring MCP**	**Little DIP**	**Little PIP**	**Little MCP**	
Extension	3.9	2.7	4.0	3.3	
Flexion	11.0	6.3	10.6	6.9	

**Table 7 sensors-24-04745-t007:** Thumb DIP flexion angle evaluation results in different viewpoints (mean error angle, unit: degrees).

Viewpoint Positions	Angular Error
Main Experiment Viewpoint	29.7°
Comparative Viewpoint 1	13.1°
Comparative Viewpoint 2	18.4°

## Data Availability

The data presented in this study are available on request.

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
