# Peer review of "An Intelligent Hand-Assisted Diagnosis System Based on Information Fusion"

_sensors, 2024, doi:10.3390/s24144745_

Round 1

Reviewer 1 Report

Comments and Suggestions for Authors

The study by Li and Zhou aims to introduce an intelligent hand assistant diagnostic system that utilizes a camera system and a Q/A approach to enhance the hand functionality of patients undergoing specific rehabilitative treatments. I particularly appreciated the authors' perspective on the topic. The literature review was satisfactory, although the addition of some papers could reinforce the significance of a multi-sensor approach and cognitive task engagement for improving hand functionality. However, I have two main concerns that could enhance the reliability of the proposed approach, which are listed below.

1)In the methods section, subsections 4.1 and 4.2, the authors did not present the complete narrative. Indeed, it is fundamental to consider the role of accurate finger joint kinematics of the hand. There are works in the literature that follow the International Society of Biomechanics to accurately reconstruct joint angle variables reliably. The standards proposed in:

"ISB recommendation on definitions of joint coordinate systems of various joints for the reporting of human joint motion—Part II: shoulder, elbow, wrist, and hand," Journal of Biomechanics 38.5 (2005): 981-992,

demonstrate a standard that merits adherence to obtain reliable clinical data. Although this seems natural, other recent papers such as:

"Evaluation of a multi-sensor Leap Motion setup for biomechanical motion capture of the hand," Journal of Biomechanics 127 (2021): 110713,

appear to only report raw comparative metrics and do not delve into the importance of adhering to standards. In my experience, glove manufacturers are particularly diligent in following these standards, validating against stereophotogrammetric systems. This may be one of the reasons behind the reduced use of Leap or camera-based approaches. I suggest the authors validate their kinematic model against a stereophotogrammetric system. Otherwise, they should discuss this aspect as a limitation of their study, but in my opinion, this is an important point to emphasize to the community.

2) The framework proposed by the authors could be particularly suitable for integrating EMG with the gamification of user experience. This study is particularly useful in highlighting the importance of integrating multiple sensors to develop technology that can significantly impact the rehabilitation field. To support this point in the discussion section, I suggest the authors review the following works:

- "Intelligent Human–Computer Interaction: Combined Wrist and Forearm Myoelectric Signals for Handwriting Recognition." Bioengineering 11.5 (2024): 458.

- "Handwritten digits recognition from sEMG: Electrodes location and feature selection." IEEE Access (2023).

Author Response

Comments 1: 

In the methods section, subsections 4.1 and 4.2, the authors did not present the complete narrative. Indeed, it is fundamental to consider the role of accurate finger joint kinematics of the hand. There are works in the literature that follow the International Society of Biomechanics to accurately reconstruct joint angle variables reliably. The standards proposed in:

"ISB recommendation on definitions of joint coordinate systems of various joints for the reporting of human joint motion—Part II: shoulder, elbow, wrist, and hand," Journal of Biomechanics 38.5 (2005): 981-992,

demonstrate a standard that merits adherence to obtain reliable clinical data. Although this seems natural, other recent papers such as:

"Evaluation of a multi-sensor Leap Motion setup for biomechanical motion capture of the hand," Journal of Biomechanics 127 (2021): 110713,

appear to only report raw comparative metrics and do not delve into the importance of adhering to standards. In my experience, glove manufacturers are particularly diligent in following these standards, validating against stereophotogrammetric systems. This may be one of the reasons behind the reduced use of Leap or camera-based approaches. I suggest the authors validate their kinematic model against a stereophotogrammetric system. Otherwise, they should discuss this aspect as a limitation of their study, but in my opinion, this is an important point to emphasize to the community.

Response 1: 

Thank you for your valuable feedback. We acknowledge the importance of adhering to ISB standards for hand kinematics. However, our markerless algorithm's performance is severely hampered by the introduction of markers, hindering accurate pose estimation. Therefore, we opted for goniometer-based validation, which, while having limitations, provides reliable joint angle measurements compatible with our approach. We have included a limitation statement in Section 4.4 acknowledging this and discussing future work on improving marker robustness.

Comments 2:

The framework proposed by the authors could be particularly suitable for integrating EMG with the gamification of user experience. This study is particularly useful in highlighting the importance of integrating multiple sensors to develop technology that can significantly impact the rehabilitation field. To support this point in the discussion section, I suggest the authors review the following works:

- "Intelligent Human–Computer Interaction: Combined Wrist and Forearm Myoelectric Signals for Handwriting Recognition." Bioengineering 11.5 (2024): 458.

- "Handwritten digits recognition from sEMG: Electrodes location and feature selection." IEEE Access (2023).

Response 2: 

Thank you for highlighting the potential of our framework for rehabilitation applications. As you pointed out, the combination of EMG with gamification of user experience has great potential. To address your suggestions, we have supplemented the relevant content in the "Wearable Sensors" section of Chapter 2.

Reviewer 2 Report

Comments and Suggestions for Authors

Dear Authors

The publication corresponds to current trends in the field of bioengineering in the use of vision systems in connection with image analysis (here, the hand, its mobility, range of motion, etc.) and the Knowledge Base Question-Answering module in the diagnosis and rehabilitation of hand dysfunctions (acquired or congenital) - so I evaluate it highly, appreciating the commitment and results achieved.

I especially appreciate the connection and construction of the Intelligent Question-Answering on knowledge graph, of course with UI.

As the authors showed in Chapters 1 and 2, many scientific and research centers, academic centers and enterprises are working on similar solutions in the field of broadly understood hand diagnostics and rehabilitation, which confirms the need to build and develop new, innovative solutions also using AI.

Assuming that the proposed solution (harware + software) - more the software part (including IQ-A, Q&A Bot), which, as I understand, is still at the development stage (the authors notice some imperfections presented in the text and summary, which I appreciate), from a substantive point of view, I evaluate the overall All right.

Further work should be focused on, among others: to cooperate with doctors of various specialties (chapter 5): orthopedics-hand surgery, nephrologists, rheumatologists, psychologists, physiotherapists and others in order to verify the currently built graph knowledge base - remember that the final diagnosis of the disease and subsequent stages of treatment should be made by the attending physician.

Comments, observations:

- I propose in Chapter 2 to systematize (classify) current solutions (described by the Authors) along with a summary of their advantages and disadvantages, I believe that it would improve the readability of the text and could potentially show the Authors the way to further search for solutions.

- what about the impact of lighting (its variability and impact on the quality of obtaining images?) when observing the hand? - it's worth mentioning.

- is there a rehabilitation module (maybe planned)? who could propose a set of rehabilitation exercises for a specific hand disability and monitor the progress of the exercises and the entire rehabilitation (of course, unless there are other diseases or contraindications) and how can/or will the doctor intervene in this case?

I consider the proposed structure of the publication: summary, division of content into chapters and their substantive content - including graphic design, as well as summary - final conclusions and literature, to be well-thought-out and reliably prepared, corresponding to the current achievements in this topic.

Author Response

Comments 1: 

- I propose in Chapter 2 to systematize (classify) current solutions (described by the Authors) along with a summary of their advantages and disadvantages, I believe that it would improve the readability of the text and could potentially show the Authors the way to further search for solutions.

Response 1: 

Thank you for the insightful suggestion to systematize the current solutions in Chapter 2. We agree that this would enhance the paper's clarity and highlight potential research avenues. We have incorporated your feedback by adding a comparative summary table (Table 1) at the end of the related work section in Chapter 2, outlining the advantages and limitations of each approach.

Comments 2:

- what about the impact of lighting (its variability and impact on the quality of obtaining images?) when observing the hand? - it's worth mentioning.

Response 2: 

Regarding the impact of lighting on image quality, we have incorporated image brightness augmentation into our algorithm to enhance robustness against varying brightness levels. Details on this approach can be found in our paper [1], ensuring our method's effectiveness under different lighting conditions.

[1] Li, H.; Chen, P.P.K.; Zhou, Y. 3D Hand Mesh Recovery from Monocular RGB in Camera Space, 2024, [arXiv:cs.CV/2405.07167].

Comments 3: 

- is there a rehabilitation module (maybe planned)? who could propose a set of rehabilitation exercises for a specific hand disability and monitor the progress of the exercises and the entire rehabilitation (of course, unless there are other diseases or contraindications) and how can/or will the doctor intervene in this case?

Response 3: 

Thank you for your insightful suggestion regarding the potential integration of a rehabilitation module. We appreciate your valuable feedback as it helps us identify areas for improvement and better align our research with practical applications.

At present, our system does not include a dedicated rehabilitation module. However, we acknowledge the significance of such a feature and have outlined our goals for further enhancement in the section (Section 4.2). Specifically, we aim to introduce additional hand assessment scales and seek collaborations with medical professionals and therapists. By working closely with domain experts, we can leverage their expertise to develop personalized rehabilitation plans tailored to individual needs and conditions.

Reviewer 3 Report

Comments and Suggestions for Authors

After review, I personally believe that the research content of this article is very novel. It studies the intelligent hand assisted diagnosis system under information fusion conditions, and the argument is also very detailed. The data can support the conclusion. It is recommended to consider publishing this article.

Author Response

Comments 1:

After review, I personally believe that the research content of this article is very novel. It studies the intelligent hand assisted diagnosis system under information fusion conditions, and the argument is also very detailed. The data can support the conclusion. It is recommended to consider publishing this article.

Response 1:

Your positive evaluation and recommendation for publishing our article are highly appreciated. We are delighted that our research on the intelligent hand-assisted diagnosis system under information fusion conditions has gained your recognition for its novelty, detailed argumentation, and data-supported conclusions.

Reviewer 4 Report

Comments and Suggestions for Authors

The paper presents an innovative approach to hand function assessment through the use of a single-vision algorithm and information fusion technology. The integration of a medical knowledge graph with a voice question-answering system is a novel concept that addresses a significant gap in the field of diagnostic tools for hand function analysis.

The paer is well-written and clear, therefore I recommend accepting the paper for publication . 

Author Response

Comments 1:

The paper presents an innovative approach to hand function assessment through the use of a single-vision algorithm and information fusion technology. The integration of a medical knowledge graph with a voice question-answering system is a novel concept that addresses a significant gap in the field of diagnostic tools for hand function analysis.

The paer is well-written and clear, therefore I recommend accepting the paper for publication .

Response 1:

We are immensely grateful for your positive evaluation and recommendation for the publication of our paper. Your recognition of our innovative approach to hand function assessment through the integration of a single-vision algorithm, information fusion technology, and a medical knowledge graph with a voice question-answering system is truly encouraging.

Round 2

Reviewer 1 Report

Comments and Suggestions for Authors

Authors addressed all my concerns.